



# First Arctic-wide assessment of SWOT swath altimetry with ICESat-2 over sea ice

Felix L. Müller, Florian Seitz, and Denise Dettmering

Technical University of Munich, Germany; TUM School of Engineering and Design, Department of Aerospace & Geodesy, Deutsches Geodätisches Forschungsinstitut (DGFI-TUM)

**Correspondence:** Felix L. Müller (felix-lucian.mueller@tum.de)

**Abstract.**

This study presents an Arctic-wide assessment of the Surface Water and Ocean Topography (SWOT) mission's swath observations of sea surface height. SWOT provides measurements in two-dimensional swaths and enables pixel-based height information with a resolution of 250 metres up to a latitudinal limit of 78°N. Although SWOT doesn't cover the central Arctic, it provides insights into SSH at an unprecedented spatial and temporal resolution. The quality of these innovative observations in such a challenging environment is evaluated through comparison with data from ICESat-2. Approximately one year of sea level anomaly data, collected between March 2023 and April 2024, is used at around 550 regionally distributed crossover locations, with measurements taken within 30 minutes. Sentinel-1 SAR imagery supports the comparisons if available. Visual comparisons of SWOT and ICESat-2 with Sentinel-1 grey-scale values reveal clear coherence. However, small-scale surface features aren't captured by SWOT as equally as by ICESat-2. The data shows absolute water level differences of about 5 cm, despite prior harmonisation of references and corrections. Differences of up to 50 cm can occur when comparing left- and right-hand SWOT swaths, mainly during winter and in areas with long sea ice coverage. This may be due to issues with the height correction from the crossover calibration. Quantitative point-by-point comparisons show mean standard deviations of about 8 cm for all surface types and 6 cm if restricted to ICESat-2-detected leads. Higher deviations are found during the early melting period between May and June, in the Canadian Archipelago and the Greenland Sea.

## 1 Introduction

The Surface Water and Ocean Topography (SWOT) mission was launched in December 2022. SWOT, an innovative altimeter mission operated by NASA and CNES, supported by the Canadian Space Agency and the UK Space Agency, has the primary objective of monitoring terrestrial water heights and the ocean surface with high spatial resolution. The unique novelty of SWOT is the use of a $K_a$-band SAR radar interferometer (KaRIn) with near-nadir incidence angles between 0.6 and 3.9 degrees. KaRIn does not sample the sea surface in nadir, but provides sea surface elevations within a 60 km left and right swath (Morrow et al., 2019). It is the first altimetry mission, which provides 2D samples of the ocean topography. The orbital configuration makes it possible to cover large parts of the Arctic (up to 77.6°N) and the entire Southern Ocean. Even though SWOT was not primarily designed as an ice mission, it opens up new possibilities for investigating the polar oceans with regard



to sea ice drift, open water detection, sea level and freeboard determination. The mission provides data with different spatial resolutions, ranging from tens of meters at the High-Rate (HR) up to approximately 500 meters (i.e., 250 meter pixel posting rate) in the Low-Rate (LR) mode and allows for the detection of ice floes and leads (i.e., elongated water openings within the sea ice) of different size.

Before using the SWOT data for scientific analyses, it should be validated in order to retrieve information on precision and
possible systematic errors. This is particularly important because of SWOT's completely new measurement principle. Since in-situ data in the Arctic is rare and not available over wide areas, one way of assessing the SWOT data quality is to compare it with data from other satellite missions, for example, with sea surface height (SSH) measurements from ICESat-2 laser altimetry.

ICESat-2, NASA's Ice, Cloud, and Land Elevation Satellite (Neumann et al., 2019) launched in 2018, carries a photon-
counting laser altimeter and is characterized by a dense observation sampling and the ability to monitor very precisely fine-scale surface changes in polar regions. ICESat-2 is primarily designed to track environmental changes in the cryosphere (e.g., ice-covered oceans, glaciers, ice sheets). Dependent on the cloud coverage, ICESat-2 samples the ocean topography in a particular dense observation pattern due to a laser beam split into 6 individual beams, a small footprint size of 13 meters, a high measurement frequency of 10 kHz (i.e. 70 cm observation point distance) and a 91-days orbit repeat cycle (Neumann et al.,
2019). Besides the monitoring of ice sheet melting, which is the major mission objective, ICESat-2 provides the opportunity to observe small-scale features of the sea ice surface, for example small leads or water openings within the sea ice cover to support sea ice freeboard or thickness computations (e.g., Kacimi and Kwok, 2022; Petty et al., 2023).

There is very little published work using SWOT in the polar regions. Armitage and Kwok (2021) used simulated SWOT data as well as precipitation measurements and $K_a$-band nadir altimetry to analyse SWOT-like backscatter histograms during sea
ice conditions. Most recently Kacimi et al. (2025) published an initial comparison of SWOT swath-altimetry with two selected overlapping ICESat-2 tracks within the Weddell and the Beaufort seas in terms of backscatter, SSH, and freeboard heights.

Our study extends the work of Kacimi et al. (2025) and presents an Arctic-wide comparison of SWOT KaRIn data with ICESat-2 laser altimetry over a longer time period, including also the science orbit phase. SWOT wide-swath altimetry is systematically evaluated in the sea ice environment and compared with ICESat-2-derived along-track sea level anomalies
(SLA) covering the entire Arctic. The comparison is based on 550 suitable crossovers between both missions with short acquisition time differences of maximum 30 minutes to minimize the impact of sea ice drift and re-freezing of leads due to rapid and abrupt temperature changes. Where possible, radar images from ESA's Corpernicus Sentinel-1A (S1) mission are additionally used as background data with pixel resolutions of 25 m or 40 m. For this purpose, dual-polarized SAR images are processed by using the SNAP (ESA Sentinel Application Platform v9.0.0 http://step.esa.int) toolbox as described in Müller
et al. (2023) to generate HH-polarized SAR images.

After a discussion of the advantages and challenges of the current SWOT data product based on selected crossover examples, we quantify the accuracy of the height measurements on the basis of all differences at the intersection points. Using statistical methods on the basis of 550 crossover spots from March 2023 until April 2024, we analyse systematic errors as well as the precision of the SWOT measurements.





## 2 Data

### 2.1 SWOT


Two different observation modes are available for SWOT in the Arctic Ocean. The low rate (LR) mode is mainly limited to ocean areas and is the predominant observation mode. The high rate (HR) mode is available in coastal and inland areas, but also seasonally between December and February in an area of about 220.000 km$^2$ in the Beaufort Sea. Both observation

modes provide pixel-based and gridded height information, but are made available as different products. In this study, sea surface height information is obtained from the Level-2 (L2) LR Unsmoothed KaRIn data set for the cal/val (1-day repeat) and science (21-days repeat) phase with an effective spatial resolution of approximately 500 m (JPL D-56407 Revision B, 2023). By default the L2 LR Unsmoothed data is provided on a spatial grid of 250 meters resolution and is made available through the NASA Physical Oceanography Distributed Active Archive Center (PO.DAAC; SWOT (2024)). In terms of available data

releases, Version C of the latest reprocessed PO.DAAC data (i.e., "PGC") is used. For nearly half of the science phase, files of the forward precessing (i.e., "PIC") are used since no reprocessed product was yet available.

SSH observations are taken from *ssh_karin_2* for left and right swath observations. These heights are already corrected by model-based delays for the dry and wet troposphere and the ionosphere, as well as for the sea state bias. Missing corrections for tides, i.e. solid Earth tides (SET), pole tides (PET) and ocean tides (OT), and the Dynamic Atmosphere Correction (DAC) are

obtained from the SWOT L2 Expert product (JPL D-56407 Revision B, 2023). The same applies to the so-called L2 calibration correction (COR; i.e. *height_cor_xover*) required to reduce systematic height errors in the sea level measurements of KaRIn. It is computed by performing a crossover analysis of ascending and descending SWOT tracks (Dibarboure et al., 2022). In summary, the SSH from SWOT is computed as followed:

$$SSH = ssh\_karin\_2 - (SET + PET + OT + DAC) + CAL \tag{1}$$

Since the corrections from L2 Expert files are sampled on a 2 km grid, they are bi-linearly interpolated to the Unsmoothed grid nodes and applied to *ssh_karin_2*. For more details on the applied corrections, we refer to the dataset documentation of L2 LR SSH Unsmoothed and Expert in JPL D-56407 Revision B (2023). In order to derive SLA, the DTU21MSS (Andersen et al., 2023) mean sea surface, available in a 1 min degree resolution, is interpolated and subtracted from the corrected SSH. In the computation, SSH data flags are taken into account to exclude data known to be of bad (i.e. invalid) quality. *ssh_karin_2*

data labelled as *bad_not_usable* and bad values of *height_cor_xover* flagged in *height_cor_xover_qual* are rejected. However, data is used that is flagged as suspect (i.e off-nominal value, but might be useful), as this flag applies to almost the entire sea-ice region (see Fig. 1).

To enable a point-wise comparison with ICESat-2, the SLA pixels from SWOT are bi-linearly interpolated to the individual ICESat-2 laser beam crossing positions for the left and the right swath, respectively. Data are removed as outliers based on

an along-beam moving standard deviation with an outlier criterion based on an interquartile range of 1.5 times above the 75% percentile and below the 25% percentile on the interpolated SWOT observations. Moreover, to minimise the effect of





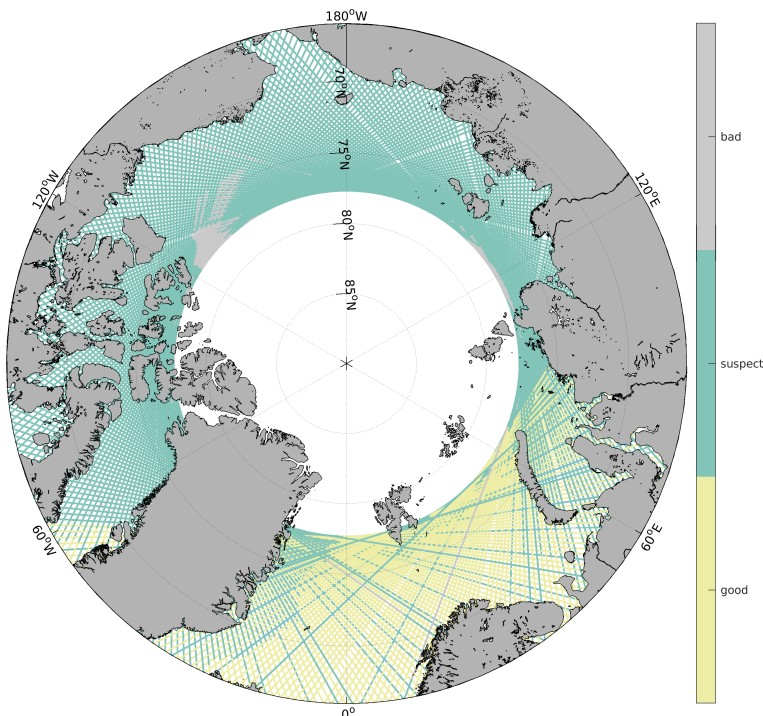

**Figure 1.** Overview of the L2 Expert quality flag of the height correction from the crossover calibration (*height_cor_xover_qual*) for January 2024 projected to the nadir altimeter position.

incorrectly increased height values at the swath edges, data within a 5 km distance to the outer and inner edges of the SWOT swaths are removed from the statistical comparisons (see Sec . 3).

## 2.2 ICESat-2

ICESat-2 latest SLA segments from ATL07 Release 006 (Kwok et al., 2023b) are used for this study. The ATL07 product covers the polar, ice-covered oceans of the Earth. It stores along-track sea level heights and anomalies of leads and sea ice for all areas with sea ice concentrations of more than 15% (Kwok et al., 2023b) for all 6 ground tracks (gt). Depending on the number of reflected photons, the spatial sample point resolution or height segment distance is variable. It ranges typically from 15 m to 30 m (Ricker et al., 2023) with a maximum of 150 m. For the comparison with SWOT KaRIn observations,

ICESat-2 elevations are utilised with and without the application of sea-ice/lead flagging in order to facilitate a comparative assessment of ATL07 heights for both surface types. However, only ICESat-2 observations from the 3 strong beams are used for statistical and quantitative analyses, because minimal elevation differences and resulting slopes between strong and weak beams are closer together than the spatial resolution of SWOT. Tests have shown that no significant differences exist between the two laser beam types.





Before comparing SLA from ICESat-2 with KaRIn swath observations, the stored ICESat-2 elevations in *height_segment_height* must undergo some pre-processing steps to allow for consistent comparisons. Therefore, the mean sea surface used specifically for ICESat-2 (MSS13; Kwok et al., 2020) is replaced by DTU21MSS (Andersen et al., 2023), and the applied dynamic inverted barometer correction (DynIB) is substituted by DAC. Furthermore, the provided sea surface elevations are converted into a mean-tide reference by adding a geoid free-to-mean conversion factor (MT_factor). As a result, the ICESat-2 SSH are

calculated as follows:

$$SSH = height\_segment\_height + MSS13 + DynIB - DAC + MT\_factor \tag{2}$$

With the exception of the introduced DTU21MSS, all corrections and conversions are included in ATL07 (Kwok et al., 2023b). See the Algorithm Theoretical Basis Document for Sea Ice Products in Appendix J (Kwok et al., 2023a) for more information on the replaced corrections.

The along-track ICESat-2 SLA are filtered by applying a standard Grubbs outlier flagging per laser beam (Grubbs, 1950). Afterwards, they are smoothed by a moving rectangle filter considering the effective SWOT Unsmoothed pixel resolution of 500 meters (JPL D-56407 Revision B, 2023).

## 2.3    Overlaps

The comparison between SWOT and ICESat-2 is performed in the Arctic Ocean between 66°N and 78°N. Crossovers between
both missions are defined when the acquisition time differs less than 30 minutes. This time interval provides a good compromise between data availability and minimization of influences on the observation scenario due to sea ice drift and rapid temperature changes (Müller et al., 2023). To search for these crossover locations, the nominal nadir altimeter ground tracks of the cal/val and science orbit of SWOT are intersected with the nominal middle ground track (i.e. gt2) of ICESat-2 over the period between 30 March and 10 July 2023 as well as 26 July 2023 and 30 April 2024. This leads to a theoretical availability of around 288
comparison spots (all laser beams combined) during cal/val and 640 during the science phase, marked with a cross in Figure 2. While most crossover locations are found in the Arctic peripheral seas, almost no comparisons are available in the eastern Greenland Sea and the Barents Sea due to a lack of sea ice coverage ($< 15\%$ sea ice concentration). Moreover, there are fewer to no usable overflights in the months of August to November due to data gaps during the transition from the SWOT cal/val phase to the scientific phase. In March 2024, there are no overlaps within 30 minutes due to the orbit configuration of ICESat-2
and SWOT. Further, the number of crossovers used is reduced by applying a minimum comparison observation number of 500 per swath (left/right) and performing the outlier rejection on the SWOT and ICESat-2 SLA as described in Sections 2.1 and 2.2 for minimizing the influence of outliers on the statistical analyses (see red dots Fig. 2). After performing the outlier rejection, observation points at 550 crossover locations remain for use in a visual and statistical comparison. The area of a crossover spot (and thus the length of the ICESat-2 ground track used for comparison) varies, since it depends on the geographical latitude
and the angle at which the two satellite missions intersect.



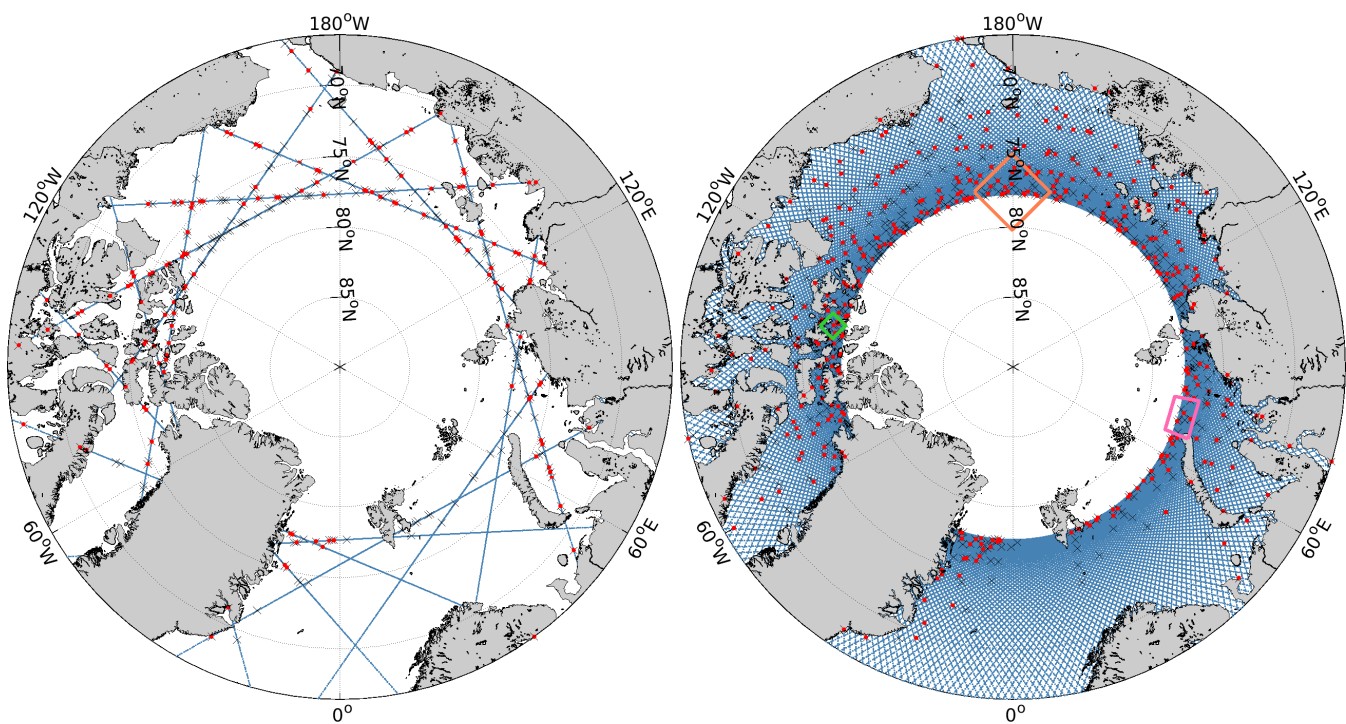

**Figure 2.** SWOT ground tracks in the Arctic (blue lines) for the SWOT cal/val phase (left) and the science phase (right). Theoretically usable crossovers between SWOT and ICESat-2 ground tracks considering land-water mask flagging and suitable cloud conditions marked with black crosses *X*, (#288 cal/val; #640 science) and effectively used crossover locations (red dots, #147 cal/val; #400 science) considering a 30-minutes time interval between both observations. Coloured rectangles show the areas of the compared examples (see following sections).

## 3 Comparison and results

Prior to the quantitative assessment of the SWOT data, visual comparisons between the three sensors of SWOT, ICESat-2 and Sentinel-1 are carried out for selected examples. This is done in order to demonstrate the potential of SWOT to represent small-scale surface types such as ice floes or leads. The SWOT SLA are compared to ICESat-2 SLA as well as to the Sentinel-1 SAR

image backscatter converted to grey-scale values. Leads or open water patches usually tend to appear very dark in SAR images (low backscatter) due to mirror-like scattering characteristics. In contrast, sea ice results in brighter grey-scale values due to a more diffuse backscattering (higher backscatter). This three-sensor comparison offers the possibility to uncover differences and similarities with regard to different surface types in addition to the direct comparison of along-track elevations.

In general, sea ice regions and detached ice floes show up in the SWOT swath data mostly as elevations in relation to

the lower lying leads and open water patches, as can be seen in Figure 3. With the exception of the swath edges, the height variability observed by ICESat-2 and SWOT shows good agreement, in particular in the three narrow lead-like structures, indicated by significant negative values in the ICESat-2 laser tracks. Deficiencies of the SWOT data become visible towards to





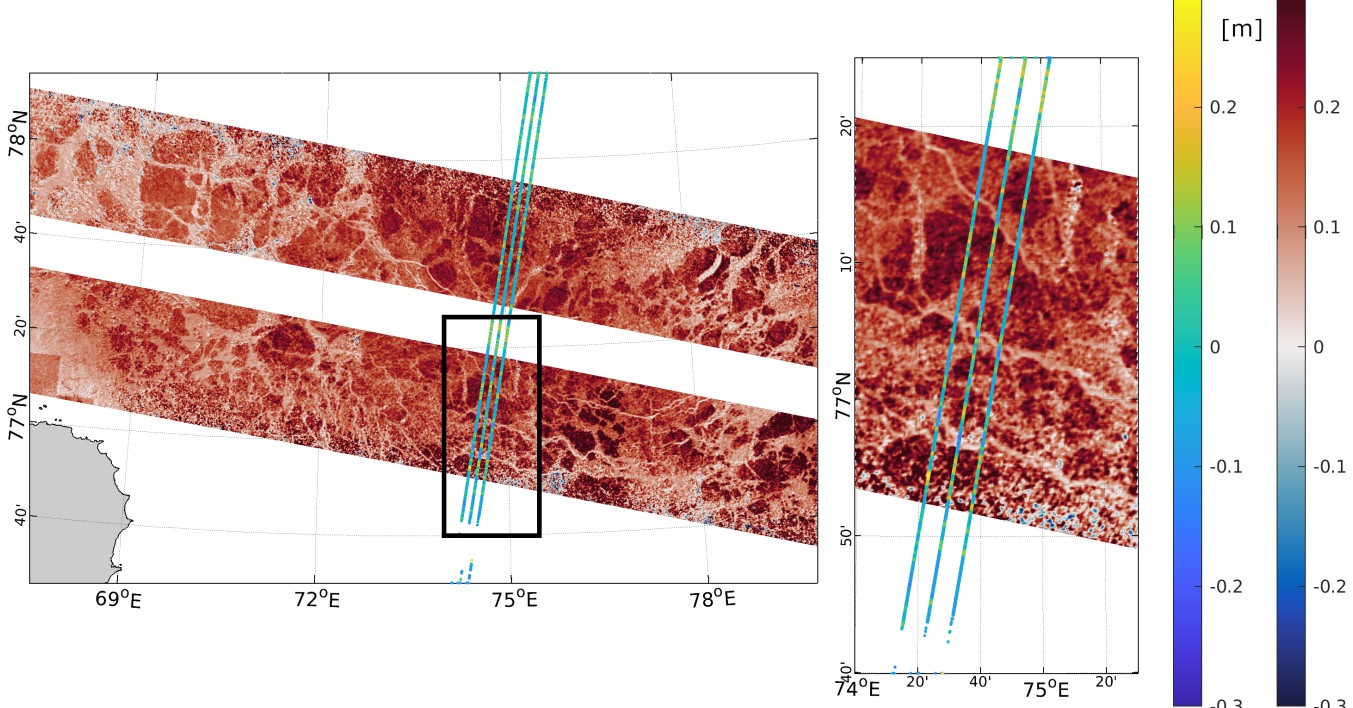

**Figure 3.** Example of SWOT swath (blue to red colorbar) and ICESat-2 (rainbow coloured) SLA observations (left) and zoom (location indicated by black box) on southern SWOT swath (right) on 2024-01-23 in the Kara Sea (see Fig. 2 pink rectangle) without subtraction of absolute mean values and the treatment of outliers at the swath edge. Acquisition time differences: SWOT vs. ICESat-2, 13 min.

the outer edges of the swath. The heights within about 5 km from the edges (both far-range and near-range) are very noisy and not reliable. These areas are thus excluded from the quantitative analysis (see Section 2.1).

Figure 4 and Figure 5 illustrate the inter-comparison between SWOT, ICESat-2 and Sentinel-1 in January 2024 in the Canadian Archipelago. While the first figure shows the geographical map, the second one displays the interpolated along-track heights. The Sentinel-1 grey-scale values show a strong coherence with SWOT and ICESat-2 SLA variations. The comparison of the 3 strong laser beams and the Sentinel-1 grey-scale values clearly reveals that the surface profile consisting of different sea ice surface types is captured quite well by SWOT. This applies in particular to distinct surface features, such as wide leads,

that are characterised by low Sentinel-1 grey-scale values or larger ice floes featured by brighter grey-scale values (e.g. in Fig. 5 highlighted in orange), which are sampled similarly to ICESat-2 in terms of height. Even smaller features, such as a lead in the right swath (zoom Fig. 4 bottom left), can be monitored very well by SWOT and ICESat-2. A closer look at an ice floe with a diameter of about 3.5 km, which is cropped in Figure 4 bottom right, shows that SWOT, similar to overlapping ICESat-2 beams, is able to detect small height differences within the ice floe surface.

However, as can be seen from Figure 4, the height determination outside the ice floe is dominated by noise, and clear elevation signatures cannot be clearly recognized or assigned to ocean topography. Further inconsistencies between elevation



**Figure 4.** Example of SWOT swath SLA, ICESat-2 profiles (in meter) and Sentinel-1 (gray-scale) image on 2024-01-25 in the Canadian Archipelago (see Fig. 2 green rectangle). Top left with SWOT, top right without SWOT. The stars indicate regions that are discussed in the text. Bottom row shows two detail views with backgrounded SAR grey-scale value information (locations are indicated with black boxes in the top right subplot). Geographical coordinates are stereographic north projected (EPSG:3413). Acquisition time differences: SWOT vs. ICESat-2, 5 min.; SWOT vs. Sentinel-1, 60 min.







**Figure 5.** Along-track height comparisons in the Canadian Archipelago (same area as in Fig. 4) for the three strong beams of ICESat-2 (blue). Interpolated SWOT heights in red and Sentinel-1 backscatter image converted to grey-scale values in grey (right axis). Orange highlighted sections show lead and floe areas; red areas show clearly dampened height variations of SWOT compared to ICESat-2. Note: Along-track height observations are zero-centred by reducing the mean per ground track (gt) and swath side.

changes from SWOT and ICESat-2 are apparent in regions with smaller ice floes close to the coasts or in the vicinity of small islands. In some regions, sea ice surfaces appear not to be represented in the SWOT heights or show no particular elevation structure compared to other sea ice surfaces. This is evident, for example, in Figure 4 in the left swath, green star, and at the

right SAR image edge, orange star, where an ice floe shows nearly no height difference relative to the surrounding water. This suggests that small-scale height variations are smoothed stronger by SWOT compared to ICESat-2, as can be seen particularly clearly in Figure 5 (bottom/middle row marked in red).




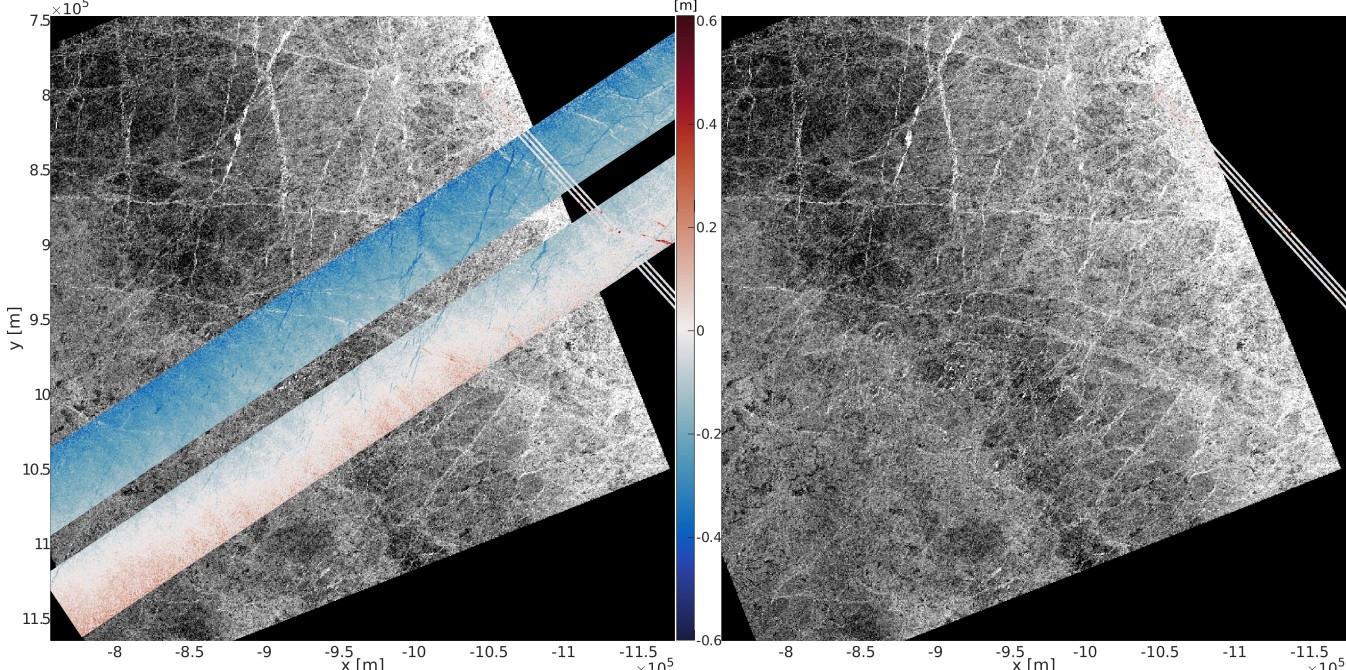

**Figure 6.** Example of coloured SWOT SLA swath, ICESat-2 profiles and a Sentinel-1 image (left) and without SWOT for comparison (right) on 2024-01-21 in the Chukchi Sea (see Fig. 2 orange rectangle). Geographical coordinates are stereographic north projected (EPSG:3413). Acquisition time differences: SWOT vs. ICESat-2, 20 min.; SWOT vs. Sentinel-1, 38 min.

A further challenge in the swath sea surface height determination arises when leads are covered by thin sea ice or frost flowers or are exposed to strong winds, roughening the water surface. In the case of the Sentinel-1 grey-scale values, such conditions result in bright line-like patterns that usually represent sea ice ridges, but can also indicate leads at the same time (Müller et al., 2023; Murashkin et al., 2018). In the SWOT elevation data, these bright patterns do not always appear to lie lower than the surroundings, but are sometimes significantly higher. This effect is particularly evident in areas where potential leads or ridges show different heights in the intersection points. Figure 6 shows an example of this effect. The area in the Chukchi Sea in January 2024 captures an almost homogeneous sea ice surface. It is only interrupted by some leads or ridges, which leave distinct height signatures in the SWOT swaths. The full explanation of what directly causes these line-like height differences remains unclear. Rather, the structure orientation (i.e., the striking) also appears to have an influence on the observed elevation differences. Additionally, this example shows large systematic cross-track errors of almost 50 cm. Probably the height correction from the crossover calibration (i.e. *height_cor_xover*) is insufficient or error-prone over almost closed sea ice areas, as no direct crossover adjustment can be conducted under these conditions (Dibarboure et al., 2022). Unfortunately, the available height calibration quality flag does not help here, since sea ice areas are entirely marked as "suspect" (see Figure 1 exemplary for January 2024). A rejection of "suspect"-labelled crossover height calibration would result in excluding all SWOT observations during sea ice conditions.





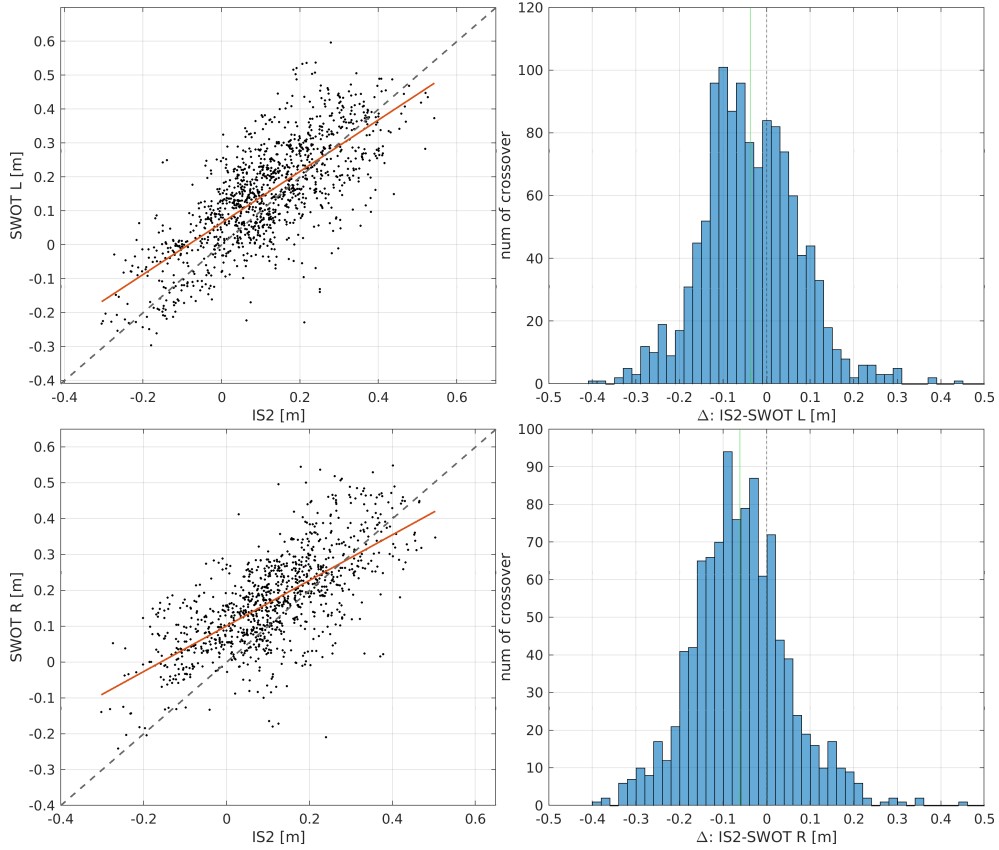

**Figure 7.** Scatter plot of ICESat-2 SLA against interpolated SWOT SLA per crossover for all 3 beams with regression line (red) and bisectrix (dashed) with histograms of mean value differences for left (top row) and right (bottom row) SWOT swaths. In the histograms, the mean values ($-4\ cm$ for left and $-6\ cm$ for right swath) are indicated by the green line, while the zero line is indicated as a dashed line.

In order to extend the assessment, a quantitative analysis of coincident SWOT and ICESat-2 sea level observations is performed. For this purpose, ICESat-2/SWOT crossovers during SWOT's cal/val and science period with a minimum of 15% sea

ice coverage (Kwok et al., 2023b) have been used.

First, mean values of the ICESat-2 laser observations and the interpolated SWOT data per left and right swath have been computed. The means are shown as scatter plots in Figure 7 for the left and for the right swath. There is generally good agreement with a correlation of about 74% for the left and 69% for the right swath. The fact that the differences are not normally distributed, but have two maxima (at least in the left swath), indicates that this is due to systematic errors in some

overflights, probably caused by incorrect cross-calibration.

Regarding the mean values of the differences, the two swaths behave quite similar and show discrepancies of $-4\ cm$ (left) and $-6\ cm$ (right), see histograms, green lines. These discrepancies are not significant given a standard deviation of around $11\ cm$ for both sides. The trend lines indicate that the mean values of SWOT tend to be systematically lower in relation to





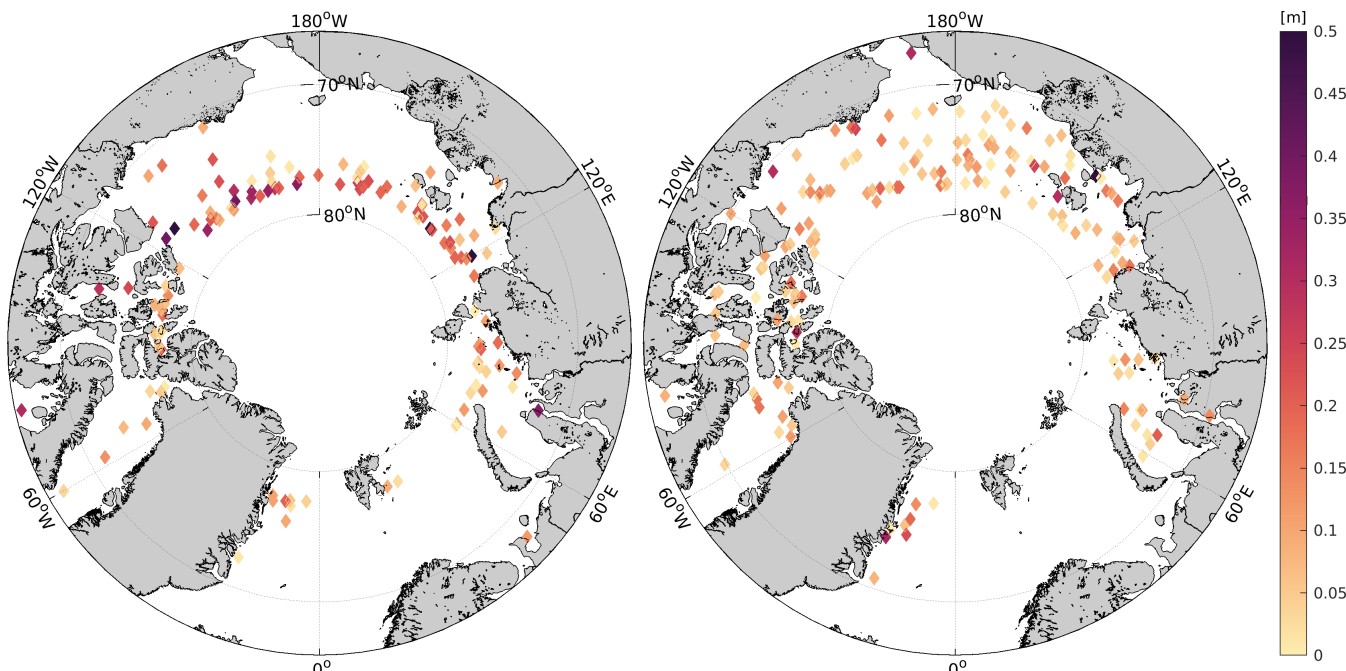

**Figure 8.** Absolute mean value differences of the interpolated SWOT observations between the left and right swath for January/February 2024 (left) and other months (right). Different laser beams are averaged per swath side. Only crossovers where the left and right swaths are available and overflown by ICESat-2 are shown.

ICESat-2. SWOT shows a smaller variation in the mean values than ICESat-2, i.e. it is less sensitive to changes in height than
ICESat-2.

The spatiotemporal differences between the left and right SWOT swaths are further studied in Figure 8. It shows the absolute differences, averaged for ICESat-2 overflights, of the mean values between the interpolated SWOT observations for the left and the right swath during the winter months January/February, which are featured by extended sea ice conditions and high crossover availability (see Fig. 9 right). It can be seen that there are generally larger deviations of up to 50 cm in Jan-
uary/February, for example in the Beaufort Sea or northern Laptev Sea. These areas are characterised by sea ice coverage of several months and high sea ice dynamics. During the other months (Fig. 8, right), the deviations are smaller and do not show a clear region-dependency.

Next, in order to analyse the height differences, the mean values are subtracted from the ICESat-2 laser profiles and the interpolated SWOT observations per swath side. Figure 9 (left) displays the histogram of the standard deviations of these
zero-centred differences. The standard deviations scatter around $0.08 \pm 0.04\,m$ and are almost normally distributed. Calculated Pearson correlation coefficients show predominantly positive and left-skewed correlations scattering around $0.4$ with maximum values up to $0.99$. With regard to seasonal variations, the lowest standard deviations are observed in autumn and at the beginning of winter between October and December, while higher standard deviations appear during the melting period from May to



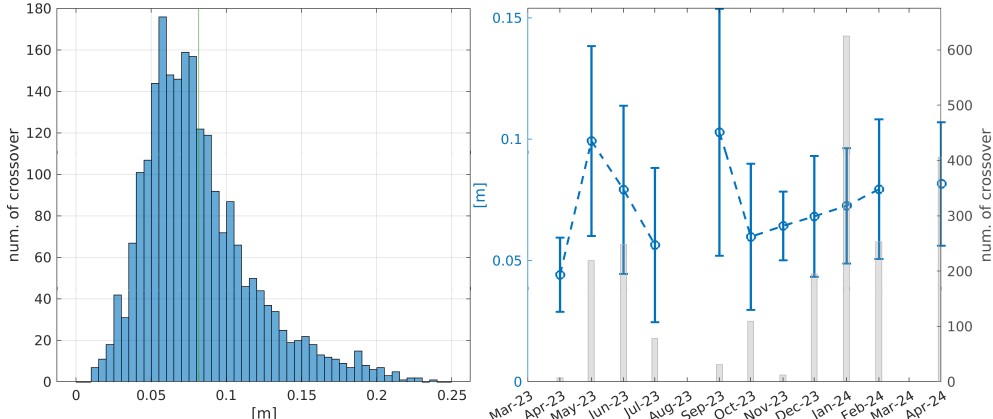

**Figure 9.** Histogram (left) and monthly averaged evolution (right) of all standard deviations of point-wise differences between interpolated SWOT swath values and strong ICESat-2 laser beams. Green line in histogram indicates mean value of $0.08\ m$

June. However, the summer months are characterised by a larger uncertainty range, as fewer crossovers are available due to the
required sea ice coverage of at least 15%.

As shown in Figure 10, the standard deviations of the crossovers are widely distributed across the Arctic Ocean region covered by SWOT. While in most areas the deviations are in the range of $4-8\ cm$, there are increased values in the area of the Canadian Archipelago, the Beaufort and Greenland Sea as well as the Northeast Greenland Shelf.

All discussed analyses refer to point-wise comparisons of observations without differentiating between observations of sea
ice and open water surfaces. To get an impression of how SWOT behaves in the case of open water spots in sea ice cover, the ICESat-2 ATL07 flag *height_segment_ssh_flag* (Kwok et al., 2023b) is applied and only filtered for open water (i.e., lead) observations detected by the ICESat-2 processing chain. This reduces the number of valid comparison points dramatically to about $180.000$ (from $11^6$) and decreases the mean standard deviation to $0.06\pm0.04\ m$, while the correlation remains unchanged.

## 4 Discussion and conclusions

The study presents an initial assessment of SWOT swath heights in the Arctic Ocean by comparing them to ICESat-2 laser altimetry heights and Sentinel-1 images. The SWOT orbit inclination does not enable a full coverage of the Arctic Ocean, but nearly 550 overlaps within a 30 minutes time frame with ICESat-2 provide the basis for visual and quantitative comparisons. Care was taken to cover as much of the sea ice period in order to observe SSH during different sea ice conditions. The SWOT data used is limited to L2 LR Unsmoothed KaRIn swath observations, as this is the predominant observation mode with
the highest spatial resolution of SWOT in the Arctic Ocean. However, meaningful sea surface heights, including all necessary geophysical corrections, can only be generated with great effort when using the SWOT Version C data by additionally including the L2 LR Expert dataset. The more recent Version D makes this problem largely obsolete, but this version is currently only available for the most recent SWOT observations.



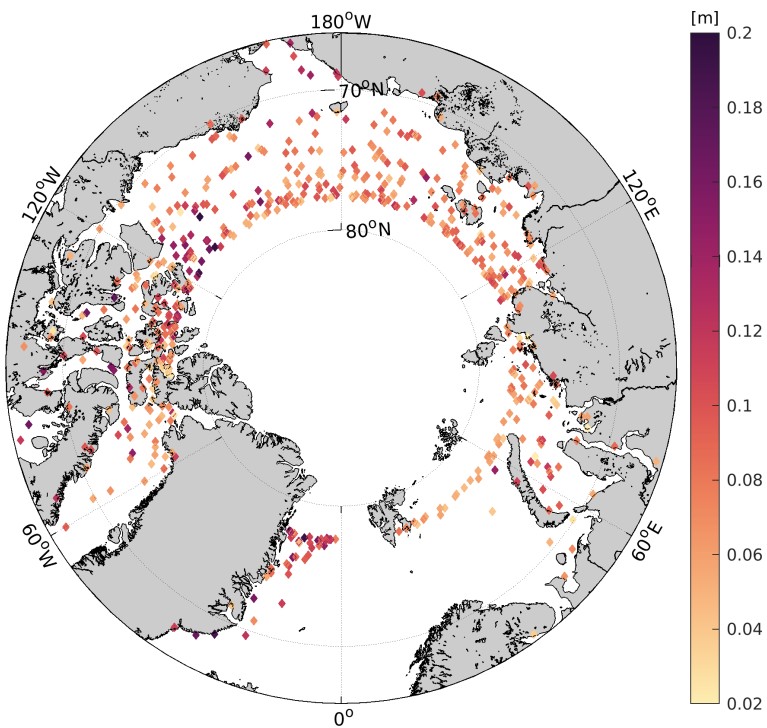

**Figure 10.** Spatial distribution of standard deviations of point-wise differences averaged per crossover (both swath sides and all three laster beams) during the study period from March-2023 until April-2024.

One critical aspect for the reliable SSH determination using SWOT is the application of the so-called height correction from
the crossover calibration. During the SWOT pre-processing (Section 2.1), bad values of this correction are removed or not used, but since alternatives are missing in the ice-covered oceans, crossover calibration corrections marked as "suspect" (Figure 1) are still applied. This results in the expectation of erroneous corrections, which can reach up to around 50 cm (Figure 6) in this study. The reason for these deviations probably lies in the way this correction is calculated for L2, as only crossovers of SWOT-SWOT overflights during open ocean conditions are used and interpolated for the observations in between. In areas of
persistent ice cover, such as the northern parts of the Arctic Ocean or the Beaufort Sea and the Laptev Sea, this can very likely lead to large uncertainties. The analysis of the data also clearly reveals that larger areas close to the edge of the SWOT swaths (approx. 5 km on each side) are very noisy and should not be used for scientific analyses.

The assessment demonstrates that SWOT has the potential to provide valuable 2D sea surface height observations during sea ice conditions. In particular, leads and open water areas, identified by their relative height differences between the sea ice sur-
face and open water patches, are clearly recognizable and respective data agree with ICESat-2 along-track height observations. In addition, SWOT is able to record small-scale elevation variations across the ice-covered surfaces, for example, of larger sea ice areas or bigger ice floes. Compared to SAR images from Sentinel-1, there is a remarkable coherence between the observed swath elevations and the interpolated grey-scale values, which allows for conclusions about different sea ice surface types (e.g.,



leads, ice floes etc.). A quantitative evaluation of the point-wise comparisons provides a mean value of the standard deviations
of the differences per laser ground track of $8 \ cm$. The precision reduces to $6 \ cm$ when only open water points are considered.
For this comparision the ICESat-2 lead flagging is used, directly taken from the ATL07 product (Kwok et al., 2023b).

With respect to small-scale features of the ice-covered sea surface, SLA from SWOT appear smoother and are characterised
by more gradual transitions between the sea ice edges and open water. In contrast, ICESat-2 profiles tend to show more
pronounced elevation variations and a clearer separation of different height or sea ice features (even if filtered to the sparser
SWOT resolution). However, it must be added here that the 532 nm wavelength of ICESat-2 naturally also ensures a lower
penetration depth and thus results in less damped reflections. Compared to SWOT, this indicates that the LR dataset can only
resolve smaller open water and sea ice areas to a limited extent. Besides wavelength differences, this can also be attributed to the
different dataset and sample resolutions. According to Kacimi et al. (2025), it is difficult for SWOT LR data to capture surface
features at shorter length scales than ICESat-2. Higher-resolution data sets of SWOT (e.g. HR, pixel-cloud) could likely reduce
this discrepancy, however, besides the resolution, other factors contribute as well. It is unclear by now how KaRIn-derived
elevations are affected by varying surface conditions, for example by snow load or thin ice. Further, it is not clear how KaRIn
responds to varying sea ice compositions and physical conditions. These questions can only be answered by detailed studies
based on validation data (in-situ or airborne).

Besides the reasons on the side of instruments and data processing, it should be mentioned that the discrepancies of the
acquisition times of the satellites add additional uncertainties due to sea ice dynamics or environmental changes, such as strong
drifts or steep temperature gradients, which can result in rapid openings and closings of leads. The 30 minutes limit applied
provides a good compromise, but a gap of 60 minutes as shown in Figure 4 between the SAR image and altimetry more
likely leads to displacements. Further challenges arise in the occurrence of roughed water surfaces, e.g., during strong-wind
conditions or the presence of frost flowers, as this leads to uncertain SSHs in the swath data. This underlines the need for
further investigations of KaRIns' backscattering characteristics and the height determination during such conditions.

Despite these uncertainties, significant new findings can be expected from SWOT in sea-ice-covered oceans. Its swath data is
a great innovation in the determination of polar sea level under the influence of sea ice and rapidly changing surface conditions.
For the first time, spatial, 2D height information for sea ice thickness or freeboard determination can be provided.

However, the study also shows that there is room for improvement in the SWOT data for generation of sea surface elevation
data under sea ice conditions. On the one hand, this concerns SWOT-dedicated corrections adapted to sea ice conditions, but
also to reliably distinguish open water from sea ice. Since currently no SWOT observation-based sea ice surface classification
exists, special focus must be given to the development of a robust classification method to detect leads. This requires a better
and deeper understanding of the $K_a$-band backscattering behaviour of KaRIn under different conditions at the sea surface
during the presence of sea ice. In this context, it is crucial to integrate information from in-situ or airborne campaigns and $K_a$-
band (i.e., SARAL) as well as laser altimetry. For example, the more detailed lead (e.g. dark/specular leads) type determination
of ICESat-2, available in ATL07, can provide further information on different lead types (e.g. dark, bright) and the evaluation
of SWOT-derived SSH within leads.



*Data availability.*

    The data used in this study is publicly available. The SWOT L2 LR Unsmoothed dataset (JPL D-56407 Revision B, 2023)
is accessed via the NASA Physical Oceanography Distributed Active Archive Center (PO.DAAC; SWOT (2024)). ICESat-2
ATL07 Release 006 (Kwok et al., 2023b) along-track data is provided via the National Snow and Ice Data Center (NSIDC).
The mean sea surface model applied DTU21MSS (Andersen et al., 2023) is taken from the DTU server (Andersen, 2025) and is
freely available. ESA Copernicus Sentinel-1A Level-1 data is made available from Alaska Satellite Facility (ASF) Distributed
Active Archive Center (DAAC). For radar image processing, the ESA Sentinel Application Platform v9.0.0 http://step.esa.int
was used, which can be obtained free of charge. The statistical analyses and the processing of the altimetry data were performed
by using MATLAB.

*Author contributions.*  FLM developed the assessment, conducted the data analysis and wrote the majority of the paper. DD contributed to
the manuscript writing and supported the study with discussions of the applied methods and results. FS supervised the research and reviewed
the manuscript.

*Competing interests.*  The authors declare no conflict of interests.

*Acknowledgements.*  This publication was supported by the Technical University of Munich (TUM).



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
