# Peer review of "First Arctic-wide assessment of SWOT swath altimetry with ICESat-2 over sea ice"

_EGUsphere, 2025_

## Author Comment (AC1)

**Response to Reviewer 2**

We thank the Reviewer for the careful and constructive comments. The suggestions and corrections have greatly improved the quality of this manuscript.

We have responded to all comments. The line numbers provided refer to the track changes document.

Review comments for "First Arctic-wide assessment of SWOT swath altimetry with ICESat-2 over sea ice" by Felix L. Müller, Florian Seitz, and Denise Dettmering

The manuscript presents a comparison of elevations measured by SWOT and ICESat-2 in the Arctic ocean (below the latitude limit set by inclination of SWOT) - most importantly in sea ice covered areas. The paper is an important step in understanding the novel altimeter measurement of SWOT over ice infested waters. The potential of SWOT is large, but we need studies like this to be able to utilise the data.

Overall, the paper is well written, interesting and important and most certainly deserves to be published. However the manuscript would benefit from a thorough, minor revision, adding more rigour and sharpening the overall presentation. The main results - overall agreement of ICESat-2 and SWOT elevations as well as the up to 50 cm differences when comparing left- and right-hand swaths are sound and scientifically the paper is of good quality. Thus, I would recommend **minor** revisions concentrating on presentation as well as adding more information on the SAR data used - including SAR / sea ice interaction as well as discussion on how snow on sea ice affects the difference between ICESat-2 and SWOT.

**More specific comments:**

Reviewer #1 has already raised a good point of using "SLA" for both elevations of sea ice and water. I'd recommend the authors to use another term, such as surface height or elevation as suggested by reviewer 1.

We modified the text and changed SLA or sea surface heights to surface elevations or heights (see comments Reviewer 1). The changes are visible in the track change document.

Also, already noted by reviewer 1, "sentinel-1 gray scale value" is a vague term. I'd recommend authors add a subsection in the data section explaining what it actually is and where does the data come from. Throughout the paper, S1 images are discussed quite a lot, but what lacks is the background on how SAR and sea ice actually interact. That is to say - there are many ways sea ice can produce high (or low) backscatter and not all of them affect the actual surface elevation of the sea ice surface.

We intentionally kept the description of the processing of Sentinel-1 SAR images to a minimum, as our study focuses on comparing SWOT and ICESAT-2. Sentinel-1 is only intended to visually support this comparison with imaging information, so the physical backscatter values were converted to an 8-bit greyscale range. For this reason, the introduction only shortly mentions the dataset used and refers to the paper by Müller et al., 2023, which contains more information about Sentinel-1 data processing and the scattering behaviour of sea ice surfaces. We have decided not to include a separate section for Sentinel-1 and to cite already existing literature (Müller et al., 2023). However, we agree that the term 'grey-scale values' is too simplistic and changed the text to be more specific how the backscatter values are handled. More information on the interpretation of SAR images are added later in the text (L156-161, see answer to first specific comment).

Another thing I'd like to see in the paper that is now lacking is the effect of snow on sea ice. Currently, snow is not really mentioned at all. Part of the mismatch between SWOT and ICESat-2 is due to penetration depth into snow (which is very shortly mentioned in L251). This should be elaborated.

Snow in connection with sea ice is a very complex topic that would deserve its own study. In the case of ICESat-2, we assume that snow is reflected at the snow surface due to the wavelength (green laser light) and that no significant penetration occurs. Therefore, we assume that the snow depth is included in the height observation and part of the sea ice freeboard (see, e.g., Petty et al., 2023).

In the case of SWOT, the influence of snow on height observations is more challenging to understand and assess, as the structure and composition of snow significantly impact the penetration depth. Moisture, temperature and composition of the snow layer in particular affect the radar scattering behaviour. In general, however, Ka-Band penetrates less deeply into the snow than the commonly used Ku-Band and tends to reflect at the upper part of the air-snow boundary (e.g. Willat et al., 2025; Fredensborg-Hansen et al., 2025; Guerreiro et al., 2016). However, in the case of KaRIn, it is still unclear to what extent the incidence angle in connection with various surface properties has an effect. Similar to nadir Ka-band altimetry, a few recent studies certify that only slight influences are to be expected (e.g. Jutila and Haas (2025), Fayne et al. (2024)). But, to gain deeper insights into KaRIn's penetration behaviour, detailed studies based on in-situ or airborne validation data are required.

With respect to the explanations above, we rephrased and restructured the text.

**Text changes L282-295:**

With respect to small-scale features of the ice-covered sea surface, surface elevations from SWOT appear smoother and are characterised by more gradual transitions between the sea ice edges and open water. In contrast, ICESat-2 profiles tend to show more pronounced elevation variations and a clearer separation of different height or sea ice features (even if filtered to the sparser SWOT resolution). This indicates that the SWOT LR dataset can resolve small open water and sea ice areas only to a limited extent. Besides wavelength differences, this can also be attributed to the different sample resolutions. Kacimi et al. (2025) also reported that it is difficult for SWOT LR data to capture surface features at shorter length scales. Higher-resolution data sets of SWOT (e.g. HR, pixel-cloud) could likely reduce this discrepancy. However, besides the resolution, other factors contribute as well. One might be the different snow penetration of both instruments.

It is common knowledge that the ICESat-2 laser data hardly show any penetration into snow or ice and represent the entire sea ice surface height (i.e. freeboard), including a possible snow layer (Petty et al., 2023). In contrast, it is still unclear how KaRIn-derived elevations are affected by varying surface conditions, for example by snow load or thin ice. In this context, there is a lack of knowledge about the sensitivity of KaRIn to different snow and sea ice properties under the influence of off-nadir incidence angles, which affect the signal propagation and backscattering behaviour. In general, Kaband penetrates less deeply into the snow than the Ku-band and tends to reflect at the upper portion of the air-snow interface (e.g. Willatt et al. (2025), Fredensborg Hansen et al. (2025), Guerreiro et al. (2016)). According to first investigations, this penetration behaviour is also to be expected for KaRIN (Jutila and Haas (2025), Fayne et al. (2024)). But in order to gain deeper insights into KaRIn's signal propagation, detailed studies based on in-situ or airborne validation data are required (Armitage and Kwok, 2021).

**Extremely specific comments:**

L140-143: The driving phenomena behind higher backscatter values is surface roughness. This should be explained and also connected to the fact that surface roughness of sea ice is very much connected to surface height. This is touched upon at  $\sim$  L170, but in my opinion quite lightly.

We agree and extend the explanations.

L156-161: In sea ice areas, the SAR image backscatter is mainly controlled by the surface roughness, but also by height and sea ice topographic variations. Leads or open water patches usually tend to appear very dark in SAR images (low backscatter) due to mirror-like scattering characteristics. In contrast, rough sea ice surfaces or topographic features like ridges or hummocks for example result in stronger backscatter due to a more diffuse backscattering (e.g. von Albedyll et al. (2024), Murashkin et al. (2018), Dierking (2013)).

L150-160 (Figures 4 and 5): The discussion here includes few wobbly statements like "grey-scale values show a strong coherence with SWOT", "can be monitored very well" and "captured quite well". These should be re-written to something preferably quantitative. In the big picture of Figure 4, one can see areas where all three datasets agree, but also areas where the correlation is *not* apparent.

Following the explanations above, the grey scale values shown in Figures 4 and 5 are not intended to represent physically quantifiable radar image backscatter or to be directly comparable with SWOT or ICESat-2 measurements. Rather, they serve as a qualitative indicator to provide a visual third reference for contextual comparison between the different datasets. For these "case studies", no quantitative comparison results are available (in contrast to the SWOT-IS2 comparison. Please also note changed x-axis (right) labels.

We added to the explanations of Figure 4 and 5 (L176-177): In this context, the SAR image is only used as a qualitative indicator to support the comparison between SWOT and ICESat-2

Figure 5: X-axes should be made equal for all subplots, even if the northernmost beam lacks data in the Western end.

The figure has been updated.

Figure 9: There is an interesting linear growth in STD over winter 23-24. This coincides with thickening ice, increasing surface roughness as well as thickening snow pack.

We agree. However, we did not further discuss this, since it is not clear whether it is related to the thickening ice or to less open water openings. This is a subject for further studies.

L190: I do not disagree, but would like to comment that pan-Arctic sea ice surface elevation distributions often have two maxima - one for first year and other for multi year ice. Would it be possible that the STD here would be a combination of two gaussian distributions, one for FYI and other for MYI?

Figure 7 shows the histogram of offsets between ICESat-2 and SWOT, not the STD. We can't exclude that the offsets depend on the ice properties, however, a relation to the quality of the cross-calibration correction is more probable, especially given the large values of the offsets.

L245: Not precision (as also pointed out by reviewer 1)

We changed that.

L260: Would it be possible to quantify the difference caused by time difference in acquisitions?

This is not only a question of the time difference but also of the speed at which the ice moves. However, it is very difficult to find values from the literature that describe sea ice movement during our investigation time period. This is particularly difficult given the regional and temporal differences in the Arctic Ocean. Therefore, based on the daily OSI SAF sea ice drift products (Global Low Resolution Sea Ice Drift - Multimission; GBL LR SID — OSI-405-c; OSI DAF (2010)), we independently determined monthly averages from November 2023 to April 2024, which show average drift velocities ranging from 280 m/h in April to 415 m/h in January 2024. In the following plot, monthly maps with maximum velocities in January up to ~620 m/h are shown:

Figure 1 Monthly sea ice drift velocities from GBL LR SID - OSI-405-c from November 2023 until April 2024.

We added the averaged values and the reference to the text (see L296 in the track change document).

Besides the reasons on the side of instruments and data processing, it should be mentioned that the discrepancies of the acquisition times of the satellites add additional uncertainties due to sea ice dynamics or environmental changes, such as strong drifts, which can reach monthly drift velocities ranging from 280±112 m/h in April 2024 until 415±154 m/h in January 2024 on average. Dependent on the season, the highest drift velocities occur in the Beaufort Sea, the Fram Strait, and the East-Greenland Current region (OSI SAF, 2010). Additionally, steep temperature gradients can cause rapid openings and closings of leads.

**References:**

Dierking, W.: Sea Ice Monitoring by Synthetic Aperture Radar, Oceanography, 26, https://doi.org/10.5670/oceanog.2013.33, 2013.

Fayne, J. V., Smith, L. C., Liao, T.-H., Pitcher, L. H., Denbina, M., Chen, A. C., Simard, M., Chen, C. W., and Williams, B. A.: Characterizing Near-Nadir and Low Incidence Ka-Band SAR Backscatter From Wet Surfaces and Diverse Land Covers, IEEE Journal of Selected Topics in Applied Earth Observations and Remote Sensing, 17, 985–1006, https://doi.org/10.1109/JSTARS.2023.3317502, 2024

Fredensborg Hansen, R. M., Skourup, H., Rinne, E., Jutila, A., Lawrence, I. R., Shepherd, A., Høyland, K. V., Li, J., Rodriguez-Morales, F., Simonsen, S. B., Wilkinson, J., Veyssiere, G., Yi, D., Forsberg, R., and Casal, T. G. D.: Multi-frequency altimetry snow depth estimates over heterogeneous snow-covered Antarctic summer sea ice – Part 1: C/S-, Ku-, and Ka-band airborne observations, The Cryosphere, 19, 4167–4192, https://doi.org/10.5194/tc-19-4167-2025, 2025.

Guerreiro, K., Fleury, S., Zakharova, E., Rémy, F., and Kouraev, A.: Potential for estimation of snow depth on Arctic sea ice from CryoSat-2 and SARAL/AltiKa missions, Remote Sensing of Environment, 186, 339–349, https://doi.org/10.1016/j.rse.2016.07.013, 2016.

Jutila, A. and Haas, C.: C and K band microwave penetration into snow on sea ice studied with off-the-shelf tank radars, Annals of Glaciology, 65, e5, https://doi.org/10.1017/aog.2023.47, 2025

OSI SAF (2010): Global Low Resolution Sea Ice Drift - Multimission, EUMETSAT SAF on Ocean and Sea Ice, DOI: 10.15770/EUM\_SAF\_OSI\_NRT\_2007, https://doi.org/10.15770/EUM\_SAF\_OSI\_NRT\_2007

Petty, A. A., Keeney, N., Cabaj, A., Kushner, P., and Bagnardi, M.: Winter Arctic sea ice thickness from ICESat-2: upgrades to freeboard and snow loading estimates and an assessment of the first three winters of data collection, The Cryosphere, 17, 127–156,380 https://doi.org/10.5194/tc-17-127-2023, 2023.

von Albedyll, L., Hendricks, S., Hutter, N., Murashkin, D., Kaleschke, L., Willmes, S., Thielke, L., Tian-Kunze, X., Spreen, G., and Haas, C.: Lead fractions from SAR-derived sea ice divergence during MOSAiC, The Cryosphere, 18, 1259–1285, https://doi.org/10.5194/tc-18-1259-2024, 2024

Willatt, R., Mallett, R., Stroeve, J., Wilkinson, J., Nandan, V., and Newman, T.: Ku- and Ka-Band Polarimetric Radar Waveforms and Snow Depth Estimation Over Multi-Year Antarctic Sea Ice in the Weddell Sea, Geophysical Research Letters, 52, e2024GL112 870, https://doi.org/https://doi.org/10.1029/2024GL112870, 2025

---

## Author Comment (AC2)

**Response to reviewer 1**

We thank the Reviewer for the careful and constructive comments. The suggestions and corrections have greatly improved the quality of this manuscript.

We have responded to all comments. The line numbers provided refer to the track changes document.

Review of "First Arctic-wide assessment of SWOT swath altimetry with ICESat-2 over sea ice" by Müller et al.

**Summary:**

This study evaluates the performance of the SWOT mission's swath-based sea surface height observations in the Arctic by comparing one year of data (March 2023–April 2024) with ICESat-2 measurements at ~550 crossover locations. Results show good overall coherence, with mean standard deviations of ~8 cm (6 cm over leads), but reveal systematic offsets between left- and right-swaths (up to 50 cm) and higher deviations during the early melt season and in complex regions such as the Canadian Archipelago.

**General Comments:**

This paper will be helpful for future studies using SWOT data over sea ice, and helps to better understand the complexity of the SWOT data. However, in its current form, it sometimes lacks clarity. Some figures are difficult to interpret and require close attention. I therefore recommend **major** revisions, although the scientific content appears sound. The main issues concern presentation and clarity."

I have some general comments that hopefully help to improve the paper:

1. The introduction lacks a clear statement of the study's main goals. It should also emphasize that SWOT is a predecessor of Sentinel-3 NG, which will carry a similar sensor, making the results valuable for the upcoming mission.

We have further emphasized our research objectives (see Introduction, L53 - L57) and added some information about Sentinel-3 Next Generation Topography (S3NG-T) in the introduction L23-27) and the conclusion (L309-311).

**These are the new paragraphs:**

L23-27: It is the first altimetry mission, which provides 2D samples of the ocean topography. Moreover, SWOT's promising observations have motivated future missions, such as the upcoming Copernicus mission Sentinel-3 Next Generation Topography (S3NG-T). This mission, planned for launch after 2030, will adopt the swath altimetry concept and bring it to an operational level. It will consist of two satellites, each carrying an across-track interferometer and a nadir-looking synthetic aperture radar (SAR) altimeter. The results of this study will help to prepare for this future mission.

L53-57: Our study presents an assessment of SWOT-KaRIn surface heights in the Arctic sea-ice regions. Due to a lack of sufficient in-situ data, a comparison with ICESat-2 laser altimetry is performed. Besides an Arctic-wide assessment, we cover a long period of time, including different seasons and both SWOT orbit phases. With this, we extend the work of Kacimi et al. (2025).

L309-311: For the first time, spatial, 2D height information for sea ice thickness or freeboard determination can be provided. We should try to learn as much as possible from the SWOT mission in order to pave the way for the future S3NG-T mission, which includes the cryosphere as a secondary objective and will extend the monitored area to a geographical latitude of 81.5°N.

2. I find the usage of SLA for heights in this paper slightly misleading. My understanding of SLA refers to open water only, not including sea ice. I would suggest to refer to surface heights/elevations instead.

We see the point and decided to use the terms **surface elevations** (instead of SLA) and **surface heights** (instead of SSH) — even if the SWOT data is also named "ssh\_karin" in the official products. Please find the changes in the track change manuscript at various locations.

3. Figures and text can be improved in clarity. For example, Figure 4 lacks labels, and the link between the different panels demands some attention. I made suggestions and comments below.

Please find our answers to this comment below.

4. Can you make a suggestion how to correct for the SWOT cross-track errors? Have you tried?

An improved cross-calibration correction is far beyond the scope of this paper. To minimize the impact of this systematic error on the comparison results, we have reduced the offset between SWOT and IS2 for both swaths individually when generating the quantitative numbers. Please find a more detailed discussion about this topic below (comment 177-179).

**Specific Comments:**

L9: Throughout the paper, it is referred to "Sentinel-1 grey-scale values". Is it related to sigma0? It might be clearer to refer directly to the backscatter coefficient.

We clarified "Sentinel-1 grey-scale values" and rephrased the sentence.

L9-10: Visual comparisons of SWOT and ICESat-2 with Sentinel-1 backscatter (i.e.  $\sigma_0$ ), converted to 8-bit grey-scale values, reveal clear coherence.

L34-42: The ICESat-2 description is maybe a bit lengthy and part of it could be moved to the Data section, especially technical details like footprint etc.

We agree and moved some ICESat-2 related information to the Data section. Please find the text in "Data", sub-section ICESat-2 in the track change document.

L106-108: ICESat-2 samples the ocean topography in a particular dense observation pattern due to a laser beam split into 6 individual beams, a small footprint size of 11 meters (Magruder et al., 2020), a high measurement frequency of 10 kHz (i.e. 70 cm observation point distance) and a 91-days orbit repeat cycle (Neumann et al., 2019).

In context of this, we also added a few lines below:

L112: Depending on the number of reflected photons, which is influenced by the surface reflectivity and cloud conditions, the spatial [...]

L38: Different values for the ICESat-2 footprint can be found in the literature; a commonly cited estimate is 11 m, as reported by Magruder et al. (2020).

We followed the suggestion and changed it to 11 m. The reference has been added. Please see the comment above.

L41: Leads are sometimes not so easy to track with ICESat-2, in contrast to radar altimetry where we receive specular waveforms. However, I would add here that ICESat-2 ice well suited to detect ridges and surface roughness (e.g., Farrell et al (2020)., Ricker et al. (2023))

In radar altimetry, leads are equally difficult to identify due to a larger footprint and possible off-nadir effects. There are numerous different methods for determining leads using radar altimeters. However, we agree with the reviewer and have added the above information including references.

**Please see in the introduction:**

L43-47: ICESat-2 aims to monitor ice sheet melting, detect ridges and provide information about surface roughness. Moreover, ICESat-2 provides the opportunity to observe small-scale features of the sea ice surface, for example small leads or water openings within the sea ice cover to support sea ice freeboard or thickness computations (Farrell et al., 2020; Kacimi and Kwok, 2022; Petty et al., 2023, Ricker et al., 2023).

L95: "ICESat-2 latest SLA segments from ATL07 Release". I believe "surface heights" are meant here? The usage of SLA is slightly confusing.

We changed SLA to "surface elevation" (see general comment 2).

L111: Here it is referred to "SSH", SLA and SSH are not the same, see also comment above. Please clarify.

Please find our comment in the "general comments" section above.

L102-104: "Tests have shown that no significant differences exist between the two laser beam types." Does it mean tests that have been done in the framework of this study? Or is it referring to earlier studies? If the latter, a reference is needed.

Initially, we created the study based on all six beams but were unable to detect any significant differences between the strong beams and the neighboring (~90-meter spacing) weaker beams. We therefore decided to use only the strong beams.

We removed the last sentence of this section.

L146: I find it hard to see the "good agreement" here. Instead of the zoom-in area, an along-track line-plot with SWOT and ICESat-2 heights might be better suited (like Figure 5). In any case, is the figure relevant? See my comments on the figures below.

The main focus of this figure is on showing general problem areas of SWOT, e.g., at the swath edges, which are removed in the later plots. Please find details on this in our comments below (reviewer comments for Figures 3 and 4). To better emphasize this, we rephrased the description of Figure 3.

L168-174: Figure 3 shows an example of a SWOT - ICESat-2 crossover in the northern Kara Sea near Novaya Zemlya Island. Sea ice surfaces and detached ice floes show up in the SWOT swath data mostly as elevations in relation to the lower lying leads and open water patches. Towards the outer edges of the swaths, especially in the last 5 km (both far-range and near-range), increased, dominant noise and erroneous surface heights become visible. These areas are thus excluded from the quantitative analysis (see Section 2.1). Additionally, artifact-like structures are observed near the

island's coastline. Besides these deficiencies, the surface elevations observed by ICESat-2 and SWOT show similar variabilities, particularly in the three narrow, lead-shaped structures (highlighted area).

L163-164: "In some regions, sea ice surfaces appear not to be represented in the SWOT heights or show no particular elevation structure compared to other sea ice surfaces" This sentence is unclear, what is meant with "sea-ice surfaces are either absent"?

We rephrased the sentence:

L191-193: In some regions, certain sea ice surfaces visible in the radar image are apparently not represented in the SWOT elevations or do not have a particular elevation signature compared to other sea ice areas.

164-165: Why are the stars not in the upper left figure? This would make the comparison easier.

Please find our answer below (reviewer comments for Figures 3 and 4).

172-177: This part is not entirely clear to me. What is meant here? What are the line-like height differences? Linear kinematic features? Perhaps it can be marked in the figure.

Our investigations in connection with Figure 6 should indicate that the observed differences in surface elevation in the regions of potential leads and ridges cannot be clearly attributed to the actual presence of these structures. We rephrased the sentence and added arrows to Figure 6.

L197-206: In the case of Sentinel-1, such conditions can cause strong backscatter (i.e., bright grey-scale values), which usually represents sea ice ridges, but can also indicate leads (Müller et al., 2023; Murashkin et al., 2018). [...] Figure 6 shows some examples of this effect (see purple arrows). In the Chukchi Sea region in January 2024, an almost homogeneous sea ice surface is visible, only interrupted by some leads or ridges, which cause distinct height signatures in the SWOT elevations. However, the observed surface elevations cannot always be unambiguously attributed to the correct surface type (i.e., clearly defined as a lead or a ridge). The full explanation of what directly causes these line-like height uncertainties remains unclear. Rather ...

177-179: this cross-track error looks like it could be at least reduced with a relatively simple gradient correction? Have you tried to correct for it?

Before we compute the standard deviations between SWOT and ICESat-2 we reduced the offsets for both swaths separately. This is reducing a large part of the error. This also holds for the along-track height plot shown in Fig. 5. It is beyond the scope of our study to develop an improved cross-calibration correction for the Arctic.

L186-189: Have you checked the behaviour using the individual ICESat-2 beams? There might be also small differences/biases between the three beams.

There are various studies that deal with offsets within the six ICESat-2 beams (e.g. Luthcke et al., 2021). However, greater attention is paid here to differences between the strong and weak beams. The resulting findings have been incorporated into the new ICESat-2 data releases (including release 6 of ATL07). Strong-only beam offsets have not yet been considered in ATL07. Studies such as Fair et al. 2024 and Smith et al., 2025 indicate systematic calibration offsets under ideal (flat, ice/snow) conditions of up to 5 cm; biases can be larger in high noise, rough, or high-scattering environments.

L194-195: "i.e. it is less sensitive to changes in height than ICESat-2" ... which makes sense given the small footprint (~11 m) of ICESat-2

We agree.

L245: "The precision reduces to 6 cm". Is it not an improvement in precision? Do you mean the standard deviation?

We changed "precision" to "mean standard deviation".

L251: "Compared to SWOT, this indicates that the LR dataset..." ... but LR is SWOT, no? Please clarify.

We removed "compared to SWOT".

Figure 3 & 4: I find Figure 4 (in combination with Fig. 5) very informative, but using the same colormap for both SWOT and ICESat-2 makes it sometimes difficult to separate. May be use different colormaps like in Figure 3? On the other hand, I wonder if Figure 3 is actually needed. Is there something in Figure 3 that cannot be explained by Figure 4 + 5? However, in Figure 4, I suggest to use "a)", "b)", etc, and moreover, it would be good to use arrows starting at the black boxes, pointing to the respective bottom figures.

Here are our responses to the various comments on Figures 3, 4 and 5:

Figure 3 (now updated due to an upload error) shows a SWOT pass without prior mean centering and application of outlier detection methods, which is the biggest difference compared to Figures 4 and 5. The image, especially on the left, shows artifacts and problems at the edges of the SWOT swaths. Since the motivation for this plot is to show the high noise in the outer edges of the swath and not a detailed along-track comparison of the heights. This is now more clearly indicated in the text (see comment above). An additional plot, similar to Fig.5, does not contain new information above the one shown in Fig. 5.

Figure 4: We agree to add labels for the different panels. However, we would like to avoid additional arrows and stars, as we believe that arrows crossing the images would overload them. We think that the rectangular markers clearly show where a larger ice floe and lead can be found. With regard to the stars in the left-hand image, we would prefer to do without additional stars, as these would obscure the effects we are describing. Furthermore, we would keep the colormaps to demonstrate how well the heights match each other. For detailed comparisons between the height profiles, Figure 5 is provided, which shows a direct along-track comparison enabling detailed analyses.

Figure 5: The right axis needs a label, even if these are relative units.

The figure is updated.

Figure 10: Colorbar label missing.

We are not sure what the reviewer means here. The unit of the colorbar "m" (meter) is specified and what is shown is described in the caption. However, we added "surface elevation".

**References:**

Farrell, S. L., Duncan, K., Buckley, E. M., Richter-Menge, J., and Li, R.: Mapping Sea Ice Surface Topography in High Fidelity With ICESat-2, Geophys. Res. Lett., 47, e2020GL090708, https://doi.org/10.1029/2020GL090708, 2020. a, b, c

Luthcke, S. B., Thomas, T. C., Pennington, T. A., Rebold, T. W., Nicholas, J. B., Rowlands, D. D., Gardner, A. S., and Bae, S.: ICESat-2 Pointing Calibration and Geolocation Performance, Earth and Space Science, 8, e2020EA001494, https://doi.org/https://doi.org/10.1029/2020EA001494, 2021.

Magruder, L. A., Brunt, K. M., and Alonzo, M.: Early ICESat-2 on-orbit Geolocation Validation Using Ground-Based Corner Cube Retro-Reflectors, Remote Sensing, 12, 3653, https://doi.org/10.3390/rs12213653, 2020.

Ricker, R., Fons, S., Jutila, A., Hutter, N., Duncan, K., Farrell, S. L., Kurtz, N. T., and Fredensborg Hansen, R. M.: Linking scales of sea ice surface topography: evaluation of ICESat-2 measurements with coincident helicopter laser scanning during MOSAiC, The Cryosphere, 17, 1411–1429, https://doi.org/10.5194/tc-17-1411-2023, 2023.

Smith, B. E., Studinger, M., Sutterley, T., Fair, Z., and Neumann, T.: Understanding biases in ICESat-2 data due to subsurface scattering using Airborne Topographic Mapper waveform data, The Cryosphere, 19, 975–995, https://doi.org/10.5194/tc-19-975-2025, 2025.